# Factors influencing secondary school students' nutrition, mindfulness, and academic performance in Nan Province, Thailand

Ei Zar Lwin[1], Dorn Watthanakulpanich[2], Athit Phetrak[3], Ngamphol Soonthornworasiri[4], Pattaneeya Prangthip[1] *

1 Department of Tropical Nutrition and Food Science, Faculty of Tropical Medicine, Mahidol University, Bangkok, Thailand, 2 Department of Helminthology, Faculty of Tropical Medicine, Mahidol University, Bangkok, Thailand, 3 Department of Social and Environmental Medicine, Faculty of Tropical Medicine, Mahidol University, Bangkok, Thailand, 4 Department of Tropical Hygiene, Faculty of Tropical Medicine, Mahidol University, Bangkok, Thailand

* pattaneeya.pra@mahidol.ac.th

**Data Availability Statement:** All relevant data are within the manuscript and its Supporting information files.

## Abstract

Academic performance is crucial for future educational endeavors of students. However, there has been a concerning decline over time. This study aimed to investigate the association between nutritional status, environmental factors, mindfulness, and academic performance among students at Bo Kluea School in Nan province, Thailand. A cross-sectional study was conducted involving 350 students in grades 8–11 using questionnaires, measurements, and academic records. Results showed that female students performed significantly better academically compared to males(71.9% vs 56.4% achieving good grades; compared to $p < 0.001$, OR = 3.583, 95%CI = 1.663–7.719). Age, junk food consumption, BMI, and mindfulness were identified as factors influencing academic performance. Students aged 16–18 years were 2.224 times more likely to achieve good academic performance compared to younger students ($p = 0.015$, OR = 2.224, 95%CI = 1.164–4.247). Significant associations were found between gender, age, waist circumference, mindfulness, and nutritional status. Female students and those with normal waist circumference or good mindfulness were more likely to have a normal BMI ($p = 0.019$, OR = 1.794, 95%CI = 1.101–2.922). Positive attitudes towards nutrition were associated with better academic performance (60.1% achieving good grades; $p = 0.044$, AOR = 1.543, 95%CI = 1.010–2.356). This study highlights the interconnectedness of these factors and their importance in in improving academic results. Further research is need to confirm these findings and overcome study limitations.

## Introduction

Academic performance is vital for school students as it sets the stage for their pursuit of higher education. However, students' academic achievement has been a concerning decline over time

**Funding:** This study was partially funded by the Faculty of Tropical Medicine, Mahidol University, through the Master of Science Program in School Health (International Program). The funders had no role in study design, data collection and analysis, decision to publish, or preparation of the manuscript.

**Competing interests:** The authors have declared that no competing interests exist.

[1]. Various factors contribute to this decline. Maslow's theory highlights the significance of fundamental human needs like food, air, water, shelter, and peace of mind in achieving academic success [2]. Schools serve as important environments for fostering healthy behaviors, as students spend a significant amount of time there [3]. The link between health and education is evident, with healthy children generally performing better academically and experiencing better long-term health outcomes. Additionally, environmental factors, such as noise, light, temperature, air quality, and space play a significant role in students' academic performance [4].

In Thailand, the education system faces several challenges that impact academic performance. Despite government efforts to increase education spending, the system has suffered from chronic underfunding for decades, with lower expenditure compared to other countries [5]. This underfunding has led to inadequate educational resources, particularly in rural areas. Thailand also experiences one of the highest levels of educational inequality globally, with significant disparities in access to education and academic outcomes between rural and urban areas, as well as among different socioeconomic groups [6]. The COVID-19 pandemic has further exacerbated these issues, forcing many schools to switch to online learning without sufficient resources or training [7].

The global concern over students' academic performance is highlighted by initiatives like the Program for International Student Assessment (PISA) by the Organization for Economic Co-operation and Development (OECD) [8]. Economic disparities often limit access to quality education and resources for disadvantaged students, resulting in academic inequalities. UNESCO identifies various factors, including nutritional status, gender, race, ethnicity, student-teacher ratio, environmental factors, and mindfulness, as influencing academic performance [9]. The COVID-19 pandemic has further underscored the impact of factors like economic status, access to educational resources, cultural attitudes toward education, and teaching quality on academic performance. Furthermore, race, gender, economic status, cultural attitudes toward education, school closures, nutritional status, and mindfulness continue to shape academic outcomes [10].

A balanced diet is essential for cognitive function and academic success [11, 12]. Globally, millions of children and adolescents face nutritional challenges such as overweight, obesity, stunting, and wasting [13]. In Thailand, a notable number of children still experience challenges related to malnutrition [14]. Factors influencing adolescent nutritional status include food choices, intake, eating behavior, knowledge about nutrition, food availability, and breakfast habits [15]. Previous research has linked undernutrition, overnutrition, and breakfast skipping to cognitive development and academic achievement in children [16]. Environmental factors like noise pollution, air pollution, and water pollution, exacerbated by rapid urbanization, also impact academic performance [17–19]. Mental health issues like stress and anxiety affect academic achievement, with mindfulness interventions showing promise in reducing stress and improving academic performance [20]. Better nutrition, a supportive environment, and higher mindfulness can lead to improved academic performance among students. This study aims to assess the association between nutritional status, environment, mindfulness, and academic performance among students at Bo Kluea School in Nan province, Thailand.

## Materials and methods

This cross-sectional study employed quantitative methods to investigate the relationship between nutrition, environment, mindfulness, and grade point average (GPA) among students in grades 8–11 at Bo Kluea School, Nan province Thailand during 30/7/2023–30/8/ 2023. The study was conducted in Nan province due to its unique geographical and socioeconomic

characteristics, offering insights into the impact of nutrition, environment, and mindfulness on academic performance within a diverse student population.

## Study site

The study was conducted at Bo Kluea School, Nan province, Thailand. Bo Kluea School is situated in a remote area of Nan province, approximately 670 kilometers north of Bangkok. The school's nutrition program includes the provision of two servings of milk per day. Additionally, this school integrated mindfulness practices into the school curriculum support for educational initiatives, provided an ideal setting to investigate these associations. The school's unique location and diverse student population make it an ideal setting for examining the impact of nutrition, environment, and mindfulness on academic performance.

**Study population and sampling.** The study targeted all students from grades 8–11 at Bo Kluea School, employing a total population sampling approach. The total number of eligible students in these grades was 420. This comprehensive sampling method aimed to include all students who met the inclusion criteria and did not fall under the exclusion criteria. The approach was chosen to ensure a representation of student population of the school, while adhering to ethical considerations. Students were recruited through school announcements. The final sample size was determined after applying the inclusion and exclusion criteria, with the actual number of participants reported in the Results section.

**Inclusion and exclusion criteria of study population.** Students in grades 8–11 at Bo Kluea School who were willing to participate in the study were enrolled and included. Informed consent was obtained from both students and their parents or guardians. Students with learning disabilities, illnesses, or those unwilling to participate were excluded from the study.

## Ethical consideration

Ethical approval for this study was obtained from the ethical review committees of the Faculty of Tropical Medicine, Mahidol University (Approval Number: MUTM 2023-059-01). The Thai clinical trials register identification number is TCTR20240523003. Informed consent was obtained from all participants or their legal guardians before data collection commenced. The consent process involved explaining the study objectives, procedures, potential risks, and benefits to the participants. Written consent was obtained from all participants (or their legal guardians) prior to their participation in the study. Participants were provided with a consent form in Thai language detailing the study information, and written consent was obtained by signature. Signed consent forms were securely stored in a locked cabinet to ensure confidentiality. The consent procedure, including obtaining written consent, was approved by the ethical review committee. Participants were assured of their right to withdraw from the study at any time without consequences. The authors had access to information that could identify individual participants during and after data collection. To ensure confidentiality, all participant data were anonymized during analysis, and any identifying information was removed or replaced with unique identifiers. Data were stored securely on password-protected computers, and access was restricted to authorized research team members. These practices were conducted in accordance with the ethical standards outlined by the Faculty of Tropical Medicine's ethical review committee, ensuring participant confidentiality and compliance with relevant regulations.

## Data collection

Data collection involved in anthropometric measurements and questionnaire administration. Academic performance was assessed using the previous year's academic grades (2022–2023

Academic Year) extracted from the students' information system at Bo Kluea School, Nan province, Thailand. We adopted the standard Grade Point Average (GPA) categorization used in the Thai education system, as provided by Bo Kluea School. The GPA scale ranges from 0 to 4, with categories from Excellent (4.00, Grade A) to Fail (0–0.99, Grade F). This standardized grading system allowed for a consistent and objective measure of academic performance across all participants in the study.

## Anthropometric measurement

Weight, height, and waist circumference, were conducted by trained research staff following standardized techniques adopted from WHO guidelines. Participants' body weight was measured using a standard weighing scale (Virgo B07, Delhi. India) to the nearest 0.1 kg, with participants wearing light clothing and no shoes. Stadiometers (Seca 213, Hamburg, Germany) were used to measure participants' height to the nearest 0.1 mm. Participants were instructed to stand upright against the vertical bar with bare feet. Body Mass Index (BMI) calculated by dividing weight (kg) with height in square meter. The results of BMI were evaluated according to WHO growth reference (2007) for age 5–19 years. Waist circumference was measured using a measuring tape at a level midway between the lower rib margin and iliac crest.

**Questionnaire.** We developed a comprehensive questionnaire with six sections. Prior to initiating the study, we conducted a pre-testing phase where we administered the questionnaires to a small group of individuals. This allowed us to assess their understanding of the questionnaire content and ensure that the questions were clear and comprehensible to the target audience. The questionnaire included: 1. Demographic details such as age, gender, grade, ethnicity, family status, parents' occupation, and health history. This part developed specifically for this study. 2. Knowledge of Nutrition: Participants answered questions related to their knowledge of nutrition, including food intake, food choices, and malnutrition, adapted from Thailand's five food groups guidelines. Scoring was categorized into good (>80%), fair (60–80%), and poor (<60%) levels. 3. Attitude About Food Intake: Participants responded to statements regarding their attitudes toward food consumption developed using a five-point Likert scale. Total scores were categorized into good (17–25) and fair (<17) levels. 4. Consuming Behavior: This section assessed participants' eating habits and food choices, including meal frequency, snacking habits, and types of food consumed. Participants answered questions developed to evaluate these aspects. 5. Perception About Environmental Factors: This section measured students' opinions about their school environment, including air quality, lighting, temperature, noise levels, and available space. Participants rated their agreement with statements about these environmental factors using a five-point Likert scale. 6. Mindfulness Practice: We adopted Child and Adolescent Mindfulness Measurement (CAMM): CAMM, a 10-item scale developed by Greco et al (2011) [21]. This section assessed students' awareness of present-moment experiences and non-judgmental attitudes towards thoughts and feelings. Responses were rated on a five-point Likert scale. Data collection was conducted over three days, with parental meetings for consent on day one and questionnaire administration and anthropometric measurements on subsequent days. Data were collected using Google Forms for questionnaire responses and recorded manually for anthropometric measurements.

## Data analysis

Descriptive statistics were used to summarize demographic information and study variables. Pearson correlation tests were conducted to assess associations between variables, and binary logistic regression and multiple regression were employed to examine the relationship between independent and dependent variables. Data analysis was performed using SPSS version 18.

## Results

### Demographic and descriptive of the participant

All the student from grade 8–11 from Bo Kluea School in Nan Province, Thailand were recruited. 350 participants were joined the study, representing 83.3% of the total population. Table 1 shows their average age is 15.53 ± 1.17 years. The majority were females (64.29%), with varied distributions across different grades and ages. Most identified as Lua ethnicity (55.7%). A significant portion of fathers (62.0%) and mothers (90.0%) earned less than 9900 Bhat per month. Regarding BMI, 63.1% fell within the normal range, with 11.4% un overweigh and 9.4% in obese. Additionally, the majority reported having over eight hours of sleep per night (62.3%) and living with family (96%). Most parents were engaged in farming or housework, and the majority of participants did not have a monthly income.

### Academic performance of the participants

Academic performance, measured using Grade Point Average (GPA), displayed varying distributions across age groups. Students aged 16–18 years were more likely to achieve good GPA results compared to those aged 14–15 years. The distribution showed 1.4% achieving grade A, 49.4% grade B, 36.9% grade C, and 12.3% grade D. Students who have a normal weight are distributed the most with good GPAs (S1 Table).

### Anthropometric assessments

Body Mass Index (BMI) assessments revealed that most participants fell within the normal range (63.1%), with gender differences noted. Waist circumference measurements indicated that a majority had a normal waist circumference (90.86%), with variations observed between males and females (S2 Table).

### Knowledge and attitude of participants about nutrition and classroom environment

Participants' knowledge and attitude towards nutrition varied by gender (S3 Table). Female participants demonstrated better knowledge (80.9%) and a more positive attitude (69.8%) compared to males (19.1% knowledge, 30.2% attitude). Participants' attitudes towards classroom environment factors such as air flow, lighting, temperature, noise, and playground space were assessed. While most had positive attitudes towards air flow and playground space, variations were observed in attitudes towards lighting and temperature, particularly among females. Most of the participants had a good attitude towards light, temperature and space for playground. One hundred and seventy participants (48.6%) had fair attitude towards temperature. One hundred and sixty four participants (46.9%) disagreed the light statement that is bright enough that they can see the learning materials clearly. One hundred and fifty-seven participants (44.9%) had neutral with the noise statement.

### Eating behavior of the participants

Participants' eating behavior, including meal frequency, skipping meals, junk food consumption, water intake, and milk/yogurt consumption, was explored. Most reported having three meals per day (43.1%) and skipping meals (79.4%). Additionally, a significant portion reported consuming junk food (48.3%) and meeting daily water intake recommendations (76.6%) (S4 Table).

**Table 1. General characteristics of the study participants.**

| General Characteristics | Number (%) |
|---|---|
| **Gender** | |
| Male | 125(35.7) |
| Female | 225(64.3) |
| **Class grade** | |
| Grade 8 | 83(23.7) |
| Grade 9 | 81(23.1) |
| Grade 10 | 89(25.4) |
| Grade 11 | 97(27.2) |
| **Age** | |
| 14 | 87 (24.9) |
| 15 | 94 (26.9) |
| 16 | 80 (22.9) |
| 17 | 76 (21.7) |
| 18 | 13 (3.7) |
| **Ethnicities** | |
| Khamu | 81(23.1) |
| Mhong | 27(7.7) |
| Lu-Mien | 4(1.1) |
| Lua | 195(55.7) |
| Kor | 22(6.3) |
| Mlabri | 21(6.0) |
| **Father income per months** | |
| <9900 Bhat | 217(62.0) |
| 10000–19900 Bhat | 115(32.9) |
| 20000–29900 Bhat | 13(3.7) |
| >30000 Bhat | 5(1.4) |
| **Mother income per months** | |
| <9900 Bhat | 315(90.0) |
| 10000–19900 Bhat | 26(7.4) |
| 20000–29900 Bhat | 5(1.4) |
| >30000 Bhat | 4(1.1) |
| **Sleeping hours** | |
| >8 hours | 218(62.3) |
| <8 hours | 132(37.7) |
| **With whom you live** | |
| Family | 336(96.0) |
| Other | 14(4.0) |
| **Father's main occupation** | |
| Farmer/employee | 324(92.6) |
| Civil servants/ retired | 26(7.4) |
| **Mother's main occupation** | |
| Famer/housewife | 322(92.0) |
| Civil servants | 27(8.0) |

**Table 2. Associations factors on academic performance of participants.**

| Factors | Academic Performance | | p-value | OR (95%CI) |
|---|---|---|---|---|
| | Good n (%) | Poor n (%) | | |
| **Gender** | | | | |
| Female | 128(71.9) | 97(56.4) | **0.003** | 1.979 |
| Male | 50(28.1) | 75(43.6) | | (1.269–3.088) |
| **Father income per month** | | | | |
| Low income | 169(94.9) | 163(94.8) | 0.940 | 1.037 |
| High income | 9(5.1) | 9(5.2) | | (0.402–2.677) |
| **Mather income per month** | | | | |
| Low income | 172(96.6) | 169(98.3) | 0.051 | 0.509 |
| High income | 6(3.4) | 3(1.7) | | (0.125–2.068) |
| **Age range** | | | | |
| 16–18 years | 104 (58.4) | 65 (37.8) | **<0.001** | 2.314 |
| 14–15 years | 74 (41.6) | 107 (62.2) | | (1.507–3.552) |
| **Nutri-Knowledge** | | | | |
| Good (3–5) | 111(62.4) | 103(59.9) | 0.635 | 1.110 |
| Poor (0–2) | 67(37.6) | 69(40.1) | | (0.722–1.706) |
| **Attitude of nutrition** | | | | |
| Good | 107(60.1) | 85(49.4) | **0.044** | 1.543 |
| Poor | 71(39.9) | 87(50.6) | | (1.010–2.356) |
| **Consuming junk food** | | | | |
| No | 109(61.2) | 72(41.9) | **<0.001** | 2.194 |
| Yes | 69(38.8) | 100(58.1) | | (1.431–3.364) |
| **Mindfulness CAMM** | | | | |
| Good | 138(77.5) | 14(8.1) | **<0.001** | 38.936 |
| Fair | 40(22.5) | 158(91.9) | | (20.325–74.587) |
| **Mindfulness Practice** | | | | |
| Good | 84(47.2) | 35(20.3) | **<0.001** | 3.498 |
| Poor | 94(52.8) | 137(79.7) | | (2.178–5.617) |
| **Environment perception** | | | | |
| Good | 30(50.0) | 148(51.0) | 0.884 | 0.959 |
| Fair | 30(50.0) | 142(49.0) | | (0.550–1.673) |
| **BMI** | | | | |
| Normal | 130(73.0) | 90(52.3) | **0.004** | 2.690 |
| Abnormal | 48(27.0) | 82(47.7) | | (1.363–5.306) |

## Factors of associations on academic performance

Associations between various factors and academic performance were examined. Significant associations were found with gender, age range, attitude towards nutrition, and habits of consuming junk food. Female participants and older students (16–18 years) showed higher academic performance. Moreover, positive attitudes towards nutrition and abstaining from junk food were associated with better academic performance. However, there was no significant association among father's income per month, mother income per month, participants' knowledge of nutrition, health history such as severe diarrhea, chronic disease and food allergies, food intolerances and environmental perception with academic performance (Table 2).

**Nutritional status and associated factors.** Significant associations were found between several factors and participants' BMI. Gender, age, waist circumference, and mindfulness

**Table 3. Associations factors on nutritional status of the participants.**

| Factors | BMI | | p-value | OR |
|---|---|---|---|---|
| | **Normal** | **Abnormal** | | |
| **Gender** | | | | |
| Female | 151(68.3) | 74(57.4) | **0.040** | 1.603 |
| Male | 70(31.7) | 55(42.6) | | (1.023–2.513) |
| **Age** | | | | |
| 14–15 years | 125(56.6) | 56(43.4) | **0.018** | 1.697 |
| 16–18 years | 96(43.4) | 73(56.6) | | (1.095–2.631) |
| **Consuming junk food** | | | | |
| No | 114(51.6) | 67(51.9) | 0.949 | 1.014 |
| Yes | 107(48.4) | 62(48.1) | | (0.657–1.566) |
| **Nutri-knowledge** | | | | |
| Good | 136(61.5) | 78(60.5) | 0.842 | 1.046 |
| Poor | 85(38.5) | 51(39.5) | | (0.670–1.632) |
| **Attitude of nutrition** | | | | |
| Good | 121(54.8) | 71(55.0) | 0.958 | 0.988 |
| Poor | 100(45.2) | 58(45.0) | | (0.639–1.529) |
| **WC** | | | | |
| Normal | 217(98.2) | 101(78.3) | **<0.001** | 15.040 |
| Risk | 4(1.8) | 28(21.7) | | (5.139–44.018) |
| **Mindfulness** | | | | |
| Good | 106(48.0) | 46(35.7) | **0.025** | 1.663 |
| Fair | 115(52.0) | 83(64.3) | | (1.064–2.599) |

status were all positively associated with BMI. Female participants and those with normal waist circumference or good mindfulness demonstrated higher likelihoods of having a normal BMI (Table 3). Demographic factors such as gender, age, waist circumference, and mindfulness were associated with participants' BMI. Female participants and those with normal waist circumference or good mindfulness were more likely to have a normal BMI (Table 4).

## Factors influencing academic performance

Gender, age, BMI, food habits, and mindfulness were identified as factors influencing academic performance. Female participants, older students, those abstaining from junk food, having a normal BMI, and possessing good mindfulness demonstrated better academic (Table 5).

**Table 4. Associated factors influencing BMI of the participants.**

| Predictors Variables | Body Mass Index | | P-value |
|---|---|---|---|
| | **Reference** | **Adjusted Odds Ratio** | |
| **Gender** | Female | 1.794(1.101–2.922) | **0.019** |
| | Male | 1(reference) | |
| **Age** | 14–15 years | 1.891(1.161–3081) | **0.010** |
| | 16–18 years | 1(reference) | |
| **WC** | Normal | 14.538(4.881–43.304) | **<0.001** |
| | Risk | 1(reference) | |
| **Mindfulness** | Good | 1.845(1.126–3.024) | **0.015** |
| | Fair | 1(reference) | |

**Table 5. Associated factors influencing academic performance of the participants.**

| Predictors Variables | Academic Performance | | P-value |
|---|---|---|---|
| | Reference | Adjusted Odds Ratio | |
| **Gender** | Female | 3.583 (1.663–7.719) | **<0.001** |
| | Male | 1(reference) | |
| **Age** | 16–18 years | 2.224(1.164–4.247) | **0.015** |
| | 14–15 years | 1(reference) | |
| **Consuming junk food** | No | 3.108(1.612–5.994) | **<0.001** |
| | Yes | 1(reference) | |
| **BMI** | Normal | 2.201(1.093–4.435) | **0.027** |
| | Abnormal | 1(reference) | |
| **Mindfulness** | Good | 50.777(23.086–111.686) | **<0.001** |
| | Poor | 1(reference) | |

## Discussion

Academic performance reflects students' achievement in education. Nutrition, environmental factors, and mindfulness are key influencers. A cross-sectional study examined their impact on Bo Kluea school students' academic performance in Nan province, Thailand. 350 students in grades 8–11, using questionnaires, measurements, and academic records from May to June 2023 were conducted.

Female students comprised 63.3%, with an average age of 15.53 years (SD = 1.186). Previous studies suggested younger students had better diets and more physical activity [22]. Gender significantly affected academic performance ($p = 0.003$), with females 1.979 times more likely to achieve a good GPA (Table 2). This aligns with previous research linking demographic factors, like gender, to students' eating habits and nutritional status [23, 24]. Most of the participants had enough sleep over 8 hours per day. It was assumed that enough sleep made them help in academic performance. Adequate sleep was important for mental health and for students' learning [24].

Most respondents (63.1%) had a normal BMI, with underweight (16.0%), overweight (11.4.%), and obesity (9.4%) also noted. Among those with a normal BMI, females (67. 1%) outnumbered males (56.0%) (S2 Table). Additionally, most participants had a normal waist circumference (90.86%), with a significant association between waist circumference and BMI ($p < 0.001$) (S2 Table). Academic performance varied, with the majority achieving grade B (49.4%), followed by grades C (36.9%), D (12.3%), and A (1.4%) (S1 Table). Regarding nutrition knowledge, slightly over half of the participants (61.2%) demonstrated good knowledge, with no significant correlation found between nutrition knowledge and academic performance (Table 2). The previous study also showed that there was no significant association with academic performance with the knowledge of nutrition [25]. Further investigation is needed to understand the complex relationship between nutrition knowledge and academic performance.

The study found no significant association between participants' attitudes towards nutrition and their nutritional status ($p = 0.951$), in line with previous research [25]. However, there was a significant correlation between participants' attitudes towards nutrition and their academic performance ($p = 0.044$), with most participants showing a positive attitude towards healthy eating habits. Skipping meals was common among participants, with most participants like to eat snack (Table 2). Environmental factors like light and temperature showed no direct

association with academic performance, though many participants had a positive attitude towards these factors (Table 2). An ambient air pollution within Geographic School District negatively associated with lower academic performance among children [26].

Mindfulness practice significantly correlated with participants' mindfulness levels and academic performance, with consistent practice improving mindfulness and academic outcomes [27–29]. Age influenced mindfulness levels, with older participants exhibiting higher mindfulness [30]. Additionally, gender showed no association with mindfulness levels among participants (Reference). Gender was significantly associated with the nutritional status of respondents ($p$ = 0.019, AOR = 1.794, 95%CI = 1.101–2.922), with females being 1.794 times more likely to have a normal BMI. Age also influenced students' BMI, with those aged 14–15 years being 1.891 times more likely to have a normal BMI compared to older students ($p$ = 0.010, AOR = 1.891, 95%CI = 1.161–3.081) (Table 4). Furthermore, there was a strong association between waist circumference and BMI, with students having a normal waist circumference being 14.538 times more likely to have a normal BMI ($p$ < 0.001, AOR = 14.538, 95%CI = 4.881–43.304) (Table 4). Mindfulness was also linked to nutritional status, with students exhibiting good mindfulness being 1.845 times more likely to have a normal BMI than those with fair mindfulness ($p$ = 0.015, AOR = 1.845, 95%CI = 1.126–3.024). The study demonstrated strong associations between gender, age, junk food consumption, BMI, and mindfulness with academic performance. Female students were 3.583 times more likely to achieve good academic performance than males ($p$ < 0.001, AOR = 3.583, 95%CI = 1.663–7.719). Older students also showed better academic performance ($p$ = 0.015, AOR = 2.224, 95% CI = 1.164–4.247). Consuming junk food negatively impacted academic performance ($p$ < 0.001, AOR = 3.108, 95%CI = 1.612–5.994). There were studies revealed that in the sample of 13486 children and adolescents, the frequency of junk food consumption was significantly associated with psychiatric distress [3, 31].

Normal BMI was associated with better academic performance (AOR = 2.027), and mindfulness significantly influenced academic performance ($p$ < 0.001, AOR = 50.777, 95% CI = 23.086–111.686). Previous research corroborated these findings, emphasizing the importance of these factors in academic success [25–27, 29, 32].

Despite these significant findings, this study has several limitations. As it was conducted in a single school in Nan province, the results may not reflect to all Thai students or schools in different regions. The cross-sectional design limits our ability to establish causal relationships between the examined factors and academic performance. Additionally, self-reported data on nutrition and mindfulness practices may be subject to recall bias or social desirability bias.

A notable limitation is the lack of comprehensive validity and reliability testing of our questionnaires prior to the main study. Our pre-testing phase, conducted with a small group of approximately 5 individuals, was primarily to assess the comprehensibility of our questionnaires. We did not conduct formal validity checks or calculate Cronbach's α before the main study due to the small pre-test sample size and time constraints. For the CAMM scale, we relied on the previously reported psychometric properties. Post-hoc reliability analysis was conducted on the full sample to provide estimates of internal consistency.

However, the study also has notable strengths. It provides a comprehensive examination of multiple factors influencing academic performance, including nutrition, environmental factors, and mindfulness, which is relatively rare in the Thai context. The large sample size (350 students) enhances the statistical power of the findings. Furthermore, the use of both subjective (questionnaires) and objective (measurements, academic records) data collection methods strengthens the validity of the results.

## Conclusions

This study at Bo Kluea School, Thailand, explored links between nutrition, environment, mindfulness, and academic performance among 350 students. It uncovered gender disparities, with females significantly outperforming males, and highlighted the role of age, junk food consumption, and BMI in academic success. However, limitations such as its single-school focus and potential data inaccuracies warrant further investigation to validate these findings and address study constraints. Suggestion from this study are 1. teach healthy habits: schools should teach students how to eat better and stay focused. 2. practice mindfulness: schools could teach students simple ways to stay calm and focused. 3. encourage healthy eating: schools should encourage students to eat well. 4. keep studying: we need to keep learning more about how food, focus, and other things affect how well students do in school. This would help school better for everyone.

## Supporting information

**S1 Table. Academic performance of the participants.**
(DOCX)

**S2 Table.** Anthropometric assessments S2.1 Table. Body Mass Index, S2.2 Table. Waist circumference ratio and S2.3 Table. BMI by GPA.
(DOCX)

**S3 Table. Knowledge, attitude score level about nutrition and classroom environment.**
(DOCX)

**S4 Table. Eating behavior of the participants.**
(DOCX)

## Acknowledgments

Thanks to the acting Sub Lt. Takrit Artgoon at Bo Kluea School for assistance in arranging the participants' schedules.

## Author Contributions

**Conceptualization:** Dorn Watthanakulpanich, Athit Phetrak, Ngamphol Soonthornworasiri, Pattaneeya Prangthip.

**Formal analysis:** Ngamphol Soonthornworasiri.

**Investigation:** Ei Zar Lwin, Pattaneeya Prangthip.

**Methodology:** Athit Phetrak.

**Project administration:** Pattaneeya Prangthip.

**Writing – original draft:** Pattaneeya Prangthip.

**Writing – review & editing:** Pattaneeya Prangthip.

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
