## [Decision Letter · Decision Letter 0]

11 Jul 2024

PONE-D-24-24017Factors Influencing Secondary School Students' Nutrition, Mindfulness, and Academic Performance in Nan Province, ThailandPLOS ONE

Dear Dr. Prangthip,

Thank you for submitting your manuscript to PLOS ONE. After careful consideration, we feel that it has merit but does not fully meet PLOS ONE’s publication criteria as it currently stands. Therefore, we invite you to submit a revised version of the manuscript that addresses the points raised during the review process.

The manuscript has received an overall positive feedback from both the reviewers. However, some corrections have been suggested which might further improve the quality of information disseminated through the article. Especially, reviewer number 1 has given detailed comments for the refinement. Please refine accordingly for further consideration.** ** ==============================

We look forward to receiving your revised manuscript.

Kind regards,

Yogesh Kumar Jain, MPH

Academic Editor

PLOS ONE

Journal Requirements:

Reviewers' comments:

Reviewer's Responses to Questions

**Comments to the Author**

1. Is the manuscript technically sound, and do the data support the conclusions?

Reviewer #1: Partly

Reviewer #2: Yes

2. Has the statistical analysis been performed appropriately and rigorously? 

Reviewer #1: Yes

Reviewer #2: Yes

3. Have the authors made all data underlying the findings in their manuscript fully available?

Reviewer #1: Yes

Reviewer #2: Yes

4. Is the manuscript presented in an intelligible fashion and written in standard English?

Reviewer #1: Yes

Reviewer #2: Yes

5. Review Comments to the Author

Reviewer #1: Thanks the authors for bringing such interesting study to the scientific community. I have forwarded my comments regarding the study as follows:-

1. The authors have described the variables and definitions of the study in their manuscript. But it is not clear whether these are adapted or adopted.

2. The manuscript lacks in defining some variables for example; Consuming Behavior, Perception about Environmental Factors and Mindfulness Practice.

3. The study also lacks in showing how the academic performance of the study participants are scored and measured?

4. Information regarding sample (sample size and sampling) should be clearly stated before the result of the study.

5. If the authors have checked the validity and reliability of the tool, there is the need to display internal validity and test-retest reliability, with the Cronbach’s α.

6. Do the authors display the crude and adjusted odds ratio in one table for the better examinations of the association between outcome variable and the predictor variables?

Reviewer #2: It is an honour to review the manuscript. This is an interesting study on “Factors Influencing Secondary School 1 Students' Nutrition, Mindfulness, and 2 Academic Performance in Nan Province, Thailand”.

My comments are appended.

Abstract:

Abstract is too long, it is better to shorten.

Introduction:

Introduction has been written in right direction. However, in my opinion it is better to put some background information or research from Thailand to draw rationale or knowledge gap.

Methods:

Methods has been written in right direction.

Results:

Results has been written in right direction.

Discussion:

Discussion has been written in right direction. The author needs to put some limitations and strength of this study.

6. PLOS authors have the option to publish the peer review history of their article (what does this mean?). If published, this will include your full peer review and any attached files.

Reviewer #1: **Yes: **Abiy Tadesse Angelo

Reviewer #2: **Yes: **Md Kamruzzaman

---

## [Author Response · Author response to Decision Letter 0]

16 Jul 2024

'Point-by-point response to editors and reviewers' 

According to comments, we agreed and corrected with all the comments raised by the editors and reviewers. We would like to take this opportunity to express our sincere thanks to editor and reviewers who identified areas of our manuscript that needed corrections or modification. We would like also to thank you for allowing us to resubmit a revised copy of the manuscript.

Reviewer #1:

The reviewer noted that while the authors described the variables and definitions of the study in their manuscript, it was not clear whether these were adapted or adopted.

Author's Response: We clarified their measures as follows:

Anthropometric measurements: Adopted from WHO standards (2007)

Nutritional knowledge: Adapted from Thailand's five food groups guidelines

Attitude About Food Intake: Self-developed using a five-point Likert scale

Consuming Behavior: Self-developed to assess eating habits and food choices

Perception About Environmental Factors: Self-developed using a five-point Likert scale

Mindfulness Practice: Adopted the Child and Adolescent Mindfulness Measurement (CAMM) by Greco et al. (2011)

Academic performance measurement: Adopted the standard GPA categorization used in the Thai education system

We revised the manuscript to clearly indicate these origins with appropriate citations.

The reviewer pointed out that the manuscript lacked definitions for some variables, such as Consuming Behavior, Perception about Environmental Factors, and Mindfulness Practice.

Author's Response: We added clearer definitions for these variables in the methodology section.

The reviewer noted that the study lacked information on how academic performance was scored and measured.

Author's Response: We added a detailed description of how academic performance was assessed, including the use of the previous year's grades and the standard Thai education system GPA categorization.

The reviewer suggested that information regarding sample size and sampling should be clearly stated before the results of the study.

Author's Response: We added a new section in the Materials and Methods describing their study population and sampling approach.

The reviewer asked about the validity and reliability of the tool, suggesting the need to display internal validity and test-retest reliability with Cronbach's α.

Author's Response: We acknowledged this limitation and provided a detailed response, including information about our pre-testing phase, tool development process, and post-hoc reliability analysis. The statement about this limitation are added in discussion section.

The reviewer asked if the authors displayed crude and adjusted odds ratios in one table for better examination of associations.

Author's Response: We presented crude odds ratios for all variables in Tables 2 and 3, and including only statistically significant predictors in the adjusted odds ratio tables (Tables 4 and 5).

Reviewer #2:

The second reviewer provided the following comments:

The abstract was deemed too long and needed shortening.

Author's Response: We revised and shortened the abstract as recommended.

The reviewer suggested adding background information or research from Thailand to the introduction to establish rationale or knowledge gap.

Author's Response: We incorporated relevant background information and research from Thailand to better establish the rationale and highlight the knowledge gap.

The Methods and Results sections were found to be written in the right direction.

For the Discussion, the reviewer suggested adding limitations and strengths of the study.

Author's Response: We added a section discussing the limitations and strengths of their study, including issues of generalizability, the cross-sectional design, potential biases, and the study's comprehensive examination of multiple factors influencing academic performance.

We would like to express that these changes would significantly improve our manuscript and thanked the reviewers for their insightful reviews.

---

## [Decision Letter · Decision Letter 1]

30 Jul 2024

Factors Influencing Secondary School Students' Nutrition, Mindfulness, and Academic Performance in Nan Province, Thailand

PONE-D-24-24017R1

Dear Dr. Prangthip,

We’re pleased to inform you that your manuscript has been judged scientifically suitable for publication and will be formally accepted for publication once it meets all outstanding technical requirements.

Kind regards,

Yogesh Kumar Jain, MPH

Academic Editor

PLOS ONE

Additional Editor Comments (optional):

Reviewers' comments:

Reviewer's Responses to Questions

**Comments to the Author**

1. If the authors have adequately addressed your comments raised in a previous round of review and you feel that this manuscript is now acceptable for publication, you may indicate that here to bypass the “Comments to the Author” section, enter your conflict of interest statement in the “Confidential to Editor” section, and submit your "Accept" recommendation.

Reviewer #1: All comments have been addressed

2. Is the manuscript technically sound, and do the data support the conclusions?

Reviewer #1: Yes

3. Has the statistical analysis been performed appropriately and rigorously? 

Reviewer #1: Yes

4. Have the authors made all data underlying the findings in their manuscript fully available?

Reviewer #1: Yes

5. Is the manuscript presented in an intelligible fashion and written in standard English?

Reviewer #1: Yes

6. Review Comments to the Author

Reviewer #1: The authors have responded to the all comments and questions that I have raised and thanks for that.

7. PLOS authors have the option to publish the peer review history of their article (what does this mean?). If published, this will include your full peer review and any attached files.

Reviewer #1: **Yes: **Abiy Tadesse Angelo

---

## [Editor Report · Acceptance letter]

2 Aug 2024

PONE-D-24-24017R1 

PLOS ONE

Dear Dr. Prangthip, 

I'm pleased to inform you that your manuscript has been deemed suitable for publication in PLOS ONE. Congratulations! Your manuscript is now being handed over to our production team.

Kind regards, 

on behalf of

Dr. Yogesh Kumar Jain 

Academic Editor

PLOS ONE